# In-N-Out: Lifting 2D Diffusion Prior for 3D Object Removal via Tuning-Free Latents Alignment

**Dongting Hu** [1]    **Huan Fu** [3]    **Jiaxian Guo** [4]    **Liuhua Peng** [1]
**Tingjin Chu** [1]    **Feng Liu** [1]    **Tongliang Liu** [2,5]    **Mingming Gong** [1,5]

[1] The University of Melbourne    [2] The University of Sydney    [3] Alibaba
[4] Google Research    [5] Mohamed bin Zayed University of Artificial Intelligence
Project Page: https://timmy11hu.github.io/3dor.github.io/

## Abstract

Neural representations for 3D scenes have made substantial advancements recently, yet object removal remains a challenging yet practical issue, due to the absence of multi-view supervision over occluded areas. Diffusion Models (DMs), trained on extensive 2D images, show diverse and high-fidelity generative capabilities in the 2D domain. However, due to not being specifically trained on 3D data, their application to multi-view data often exacerbates inconsistency, hence impacting the overall quality of the 3D output. To address these issues, we introduce "In-N-Out", a novel approach that begins by inpainting a prior, i.e., the occluded area from a single view using DMs, followed by outstretching it to create multi-view inpaintings via latents alignments. Our analysis identifies that the variability in DMs' outputs mainly arises from initially sampled latents and intermediate latents predicted in the denoising process. We explicitly align of **initial** latents using a Neural Radiance Field (NeRF) to establish a consistent foundational structure in the inpainted area, complemented by an implicit alignment of **intermediate** latents through cross-view attention during the denoising phases, enhancing appearance consistency across views. To further enhance rendering results, we apply a patch-based hybrid loss to optimize NeRF. We demonstrate that our techniques effectively mitigate the challenges posed by inconsistencies in DMs and substantially improve the fidelity and coherence of inpainted 3D representations.

## 1 Introduction

Neural Radiance Fields (NeRFs) [50, 2, 23, 58, 81, 37, 13, 10, 3, 92] have effectively revolutionized 3D scene reconstruction from multi-view images. These models offer high-fidelity novel-view synthesis, proving beneficial across a variety of domains [32, 87, 88, 43, 107, 6, 63, 72, 8, 60]. Despite the impressive ability to reconstruct highly detailed scenes, these learning-based methods depend on the availability of consistent multi-view training data. This reliance limits their generalizability, particularly in editing 3D representations for tasks like object removal and inpainting occluded areas.

Recently, diffusion models (DMs) [30, 18, 71, 80] have gained significant attention in the field of generative modelling for 2D images. These models are well-known for their robustness as generative priors, capable of producing diverse and high-fidelity results in 2D inpainting tasks. However, adapting these 2D priors for 3D object removal is not straightforward. While the inherent diversity of DMs benefits the generation of varied outputs, it also poses a significant challenge: high variance in the inpainted results (Fig.1 middle column). Consequently, these models frequently produce outputs that, while visually appealing in isolation, may appear misaligned when incorporated into 3D domain [53, 94, 93, 25, 19, 97, 84]. This misalignment often results in the loss of high-frequency details, crucial for realistic and coherent scene rendering.

38th Conference on Neural Information Processing Systems (NeurIPS 2024).

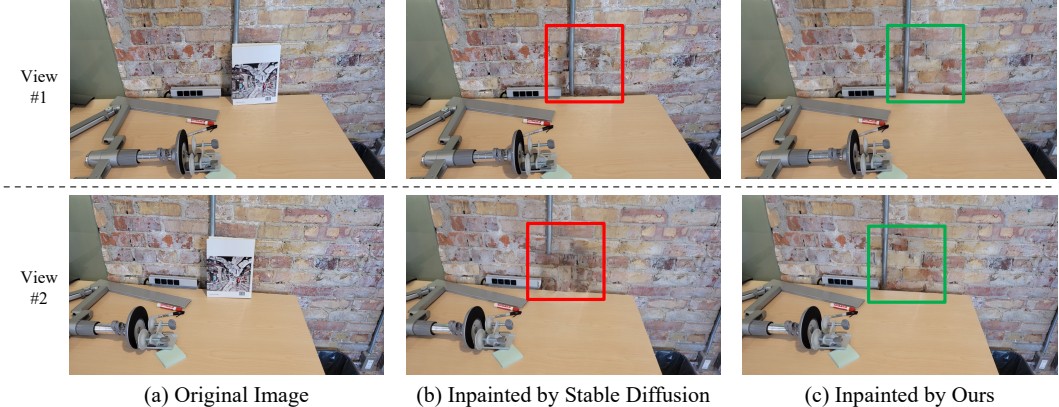

| (a) Original Image | (b) Inpainted by Stable Diffusion | (c) Inpainted by Ours |

Figure 1: Inpainting outcomes of multi-view images from original Stable Diffuion [71] (middle) with those achieved by our approach (right). The inpainted areas are highlighted in red and green boxes.

Previous studies addressing such 3D inconsistencies can be broadly categorized into two approaches: multi-view and single-view priors. The former tackled inconsistencies across multi-view inpainted images by optimizing NeRFs with modified objectives [53, 94, 93, 65, 91]. While these methods have shown promise in refining inconsistent inputs, they sometimes suffer from a loss of detail fidelity during the training process, as illustrated in Fig. 4. Conversely, other studies have attempted to overcome the multi-view inconsistency bottleneck by anchoring the inpainting process to a single reference image that serves the entire scene [42, 51, 109]. This approach, however, places significant reliance on the selection of an appropriate reference image and the accuracy of depth estimates, which could lead to geometric artifacts during testing, as shown in Fig. 4.

To address these challenges, we aim to overcome 3D inconsistencies by guiding 2D DMs to achieve multi-view consistent inpainting results (Fig. 1 right column). Our analysis reveals that the variance in model outputs primarily comes from the random noise as the **initial** latent sample, and **intermediate** latents inferred by the denoising network. Each frame's initial latents are independently sampled, while intermediate latents are individually predicted, highlighting how view-dependent data impacts the generation process. Therefore, our approach focuses on aligning these two critical elements across multiple inputs. We introduce "In-N-Out", a conditional-sampling-like approach that inpaints a sampled view and outstretches it to multiple views. Our method contains three key components:

1. Conditional Inpainting Pipeline: We propose a pipeline that first samples an inpainting outcome from a random view as an inpainting prior. This prior then serves as a condition to guide the inpainting process of multiple views, ensuring a consistent inpainting foundation.

2. Explicit Latents Alignment: Leveraging the geometry derived from a pre-trained NeRF and the inpainting prior, we sample multi-view initial latents conditional on the geometry dictated by the inpainting prior. This ensures that the primary components within the inpainted areas are structurally consistent and align with the underlying 3D geometry.

3. Implicit Latents Alignment: We employ a cross-view attention mechanism during the denoising steps to align predicted intermediate latents concerning the inpainting prior. This enhances the appearance consistency across the inpainted images.

To further enhance our method's performance in the 3D domain, we have implemented a patch-based optimization strategy using a hybrid loss on our inpainted multi-view images. This strategy employs perceptual loss to rectify spatial mismatches, and adversarial loss to preserve high-frequency details. By addressing these key challenges, our framework effectively handles multi-view inconsistencies and enhances the fidelity and coherence of 3D representations. The effectiveness of our approach is demonstrated through both qualitative and quantitative evaluations of a challenging object removal dataset. Our results indicate comprehensive improvement compared to existing methods, highlighting our model's ability to achieve greater fidelity and consistency in inpainted scenes.

# 2 Related Works

**2D Editing with Diffusion Models** Diffusion models [30, 33, 105, 59, 79, 80], have revolutionized image generation with their capacity to create highly realistic images. These models facilitate customizable generation via textual prompts [18, 29, 69, 74], predominantly using pre-trained Stable Diffusion [71]. Several editing methods [22, 57, 56, 61] allow users to adjust images by moving anchor points to new locations. Editing typically begins by inverting the latent representation of the image to be edited back to its initial noise [80], with modifications made during the denoising phase. Prompt-to-Prompt (P2P) [26] edits images by adjusting the cross-attention between the image and text. Null-text inversion [55] addresses artifacts in DDIM inversion [80] when using classifier-free guidance [29]. Delta Denoising Score (DDS) [27] optimizes the latent image representation by aligning the predicted noises of the original and modified texts. Additionally, several studies [7, 9, 26] have identified a relationship between the appearance of images generated by diffusion models and the key-value pairs. While these advancements represent significant progress preserve some content from the original image in 2D image editing, they do not account for multi-view consistency, thus can not be lifted to 3D editing directly.

**Lifting 2D diffusion models for 3D editing** Recent advancements in 3D editing and generation have effectively utilized 2D DMs to enhance these processes, as demonstrated in various studies [52, 76, 96, 62, 97, 99, 41, 95, 68, 49, 103]. Pioneering works have used images inferred by DMs for direct supervision. Instruct-NeRF2NeRF (IN2N) [25] approached the editing task by transforming 3D model editing into a 2D image editing task, utilizing Instruct Pix2Pix (IP2P) [5] to iteratively update 3D scenes. Similarly, ViCA-NeRF [19] addressed editing challenges by modifying reference images and integrating these changes into the scene. DreamEditor [110] opted for a different strategy by converting NeRF into a mesh for direct optimization. GaussianEditor [12] applies semantic tracing to identify and modify editing targets within 3D Gaussian Splatting (3DGS) [37]. Similarly, Gaussian Grouping [100] implements Identity Encoding for each Gaussian to create masks for editing. Conversely, Score Distillation Sampling (SDS) [64] provides an alternative way to guide 3D representations by backpropagating gradients from a diffusion model's denoiser [1, 16, 73, 71] into the underlying scene representation. This technique has been effectively applied to generate realistic 3D and 4D scenes using NeRFs [11, 40, 39, 108, 110] and 3DGS [70, 101, 84, 15].

**2D and 3D Inpainting** 2D inpainting methods reconstruct images by filling missing content in areas defined by a mask [20, 77, 104, 86, 46, 75, 102]. Early techniques, exemplified by [21], relied on copying textures from known to unknown regions. LaMa [82] excels in restoring large missing areas using fast Fourier convolutions, extensive receptive fields, and large training masks. Although highly effective at generating plausible background textures within specified masks, LaMa limits the fidelity of its outputs. In contrast, probabilistic diffusion models [30] have shown impressive results in image generation and offer a wide range of inpainted outputs. DMs can be adapted for inpainting without specific training, and modify known regions during each denoising step to fit the task [47]. Similarly, Stable Diffusion [71] excels at inpainting by operating within latent space, allowing for efficient and effective image generation. In this work, we adopt it as our 2D inpainter.

3D scene inpainting aims to fill missing areas within a space, such as removing objects and generating coherent geometry and textures to complete the scene. Although 3D generative models have garnered large interest [4, 34, 38, 89, 78, 44, 31, 85, 45, 14, 12], they are often limited by the scarcity of 3D training data, hence result in poor generalization, particularly in scene inpainting tasks. Therefore, most current 3D inpainting models [53, 51, 42, 65, 93, 94, 109, 91] enhance their effectiveness by adopting priors from 2D models. SPIn-NeRF [53] reduces multi-view inconsistencies by first inpainting views and then optimizing NeRF using perceptual loss. NeRFiller [93] tackles multiple frames simultaneously by tiling images for DMs. GaussianEditor [12] edit targets within 3DGS [37], guided by inpainted multi-view images from DMs. While these methods show promise, they can sometimes compromise detail fidelity during training. Alternatively, some studies circumvent multi-view inconsistencies by using a single reference image for the entire scene [42, 51, 109]. Infusion [109] stands out in the inpainting of 3DGS, leveraging a pre-trained depth completion network to infer point clouds from a single inpainted view, though this method depends heavily on precise depth estimates. Concurrent works [65, 54] address these challenges using SDS objective [64] to better align 2D model priors with 3D scene consistency.

# 3 Preliminaries

## 3.1 Neural Radiance Fields

Neural Radiance Fields (NeRFs) [50] represents a breakthrough in 3D rendering by employing a multilayer perceptron (MLP), denoted as $\phi$ to represent a scene. This MLP serves as a continuous volumetric function to capture and reconstruct a scene in unprecedented detail. Specifically, NeRFs take as input the view direction $d$ and a 3D coordinate $r(\tau)$ sampled from a camera ray defined by $r(\tau) = o + \tau d$. At each position along this ray $r(\tau)$, the network predicts the volume density and view-dependent color, represented as $(\sigma, c)$. To render a camera pixel, NeRFs perform an aggregation of the predicted densities and color emissions $\sigma(\tau_i), c(\tau_i)$ along the camera ray. This process is mathematically formulated as an approximation of a volume rendering integral [48], which is used to compute the final color of the pixel:

$$\hat{C}(r) = \sum \Gamma_i \left(1 - \exp\left(-\sigma(\tau_i)\delta(\tau_i)\right)\right) c(\tau_i), \quad \text{with } \Gamma_i = \exp\left(-\sum\nolimits_{j=1}^{i-1} \sigma(\tau_j)\delta_j\right), \quad (1)$$

where $\delta(\tau_i) = \tau_{i+1} - \tau_i$ is the distance between adjacent samples along the ray. During the training phase, rays are uniformly sampled from the training images, and the volumetric field is optimized using mean square error (MSE) to enhance the accuracy and realism of the rendered scenes.

## 3.2 Diffusion Models

Diffusion models [30] consist of two processes: a forward process that gradually introduces noise to a data sample $z^0 \sim p_{\text{data}}(z)$, and a learned reverse process that iteratively denoises a purely Gaussian noise sample $z^T \sim \mathcal{N}(0, 1)$ back into a clean image $z^0$. The reverse process is parameterized by a conditional noise prediction network $\epsilon_\theta$, trained to predict the noise using the simplified objective:

$$p_\theta(z^{0:T}|c) = p(z^T) \prod_{t=1}^{T} p_\theta(z^{t-1}|z^t, c), \quad p_\theta(z^{t-1}|z^t, c) = \mathcal{N}(z^{t-1}; \mu_\theta(z^t, t, c), \sigma^2 I), \quad (2)$$

where $t$ is the time step in the diffusion process, $z^t$ is an intermediate noisy sample, and $c$ represents a condition (e.g., images, masks, or text). Utilizing a deterministic sampler like DDIM [80], the sample $z^{t-1}$ can be obtained by $z^{t-1} = z^t - \epsilon_\theta(z^t, t, c)$; note that scaling is omitted for simplicity. In practice, as we use Stable Diffusion [71], a latent diffusion as the inpainting backbone, $z$ is latent and the generated image is obtained with a decoder $\Omega(z^0)$. Hence, the variability of the generated image $z^0$ depends solely on initial **sampled** latent $z^T$ and intermediate **inferred** latents $\left\{z^{t-1}\right\}_{t=1}^{T}$.

# 4 Method

Given a set of multi-view training images $\{\mathcal{I}_i\}_{i=1}^{N}$ from the scene with corresponding masks $\{\mathcal{M}_i\}_{i=1}^{N}$ indicate the unwanted object in each frame, our approach seeks to generate consistently inpainted training set $\{\tilde{\mathcal{I}}_i\}_{i=1}^{N}$ and use them to supervise NeRF. Our approach is structured into three key stages:

- Stage 1: Pretrain a NeRF $\phi$ using $\{\mathcal{I}_i\}_{i=1}^{N}$ and $\{\mathcal{M}_i\}_{i=1}^{N}$, along with a sampled inpainted prior $\tilde{\mathcal{I}}_p$ as a rough hallucination of the inpaint feature. (Sec. 4.1).
- Stage 2: Leverage $\phi$ to inpaint additional views $\{\tilde{\mathcal{I}}_i \mid i \neq p, i = 1, \dots, N\}$ conditioned on the inpainting prior $\tilde{\mathcal{I}}_p$ via explicit and implicit latents alignment. (Sec. 4.2)
- Stage 3: Using the inpainted image set $\{\tilde{\mathcal{I}}_i\}_{i=1}^{N}$, we optimize $\phi$ with a patch-based hybrid loss to distill multi-view supervision. (Sec. 4.3)

An overview of our method is shown in Fig. 2.

## 4.1 Stage 1: Pre-train NeRF

The initial stage involves training the NeRF on the unmasked region, we follow the original work [50] where simple MSE loss is applied:

$$\mathcal{L}_{\text{rec}}(\phi) = \sum_{r \in R_{\text{unmasked}}} \left\| \hat{C}_\phi(r) - C(r) \right\|_2^2, \quad (3)$$

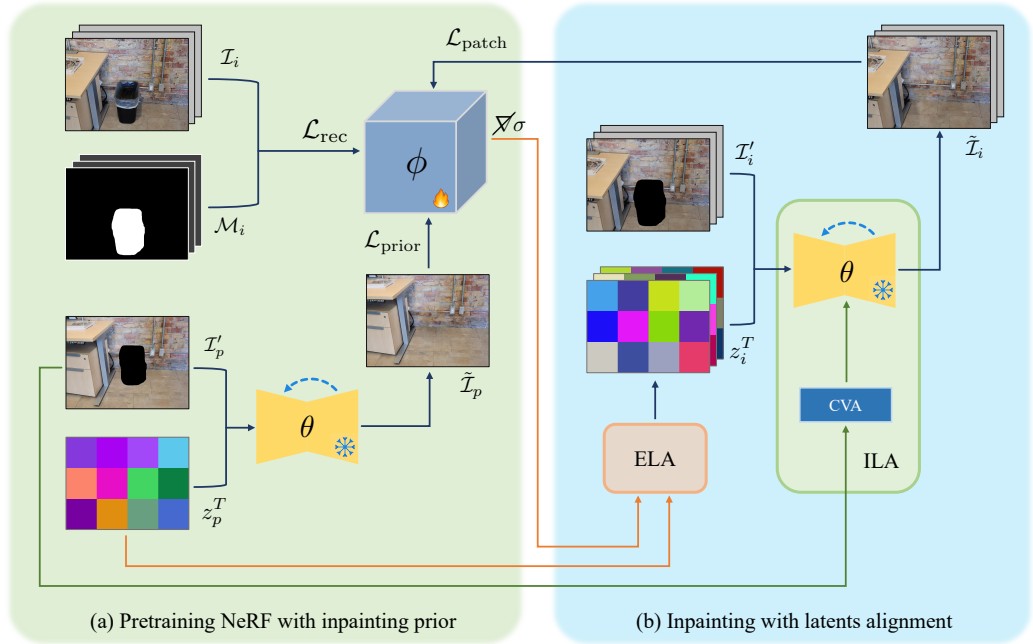

(a) Pretraining NeRF with inpainting prior      (b) Inpainting with latents alignment

Figure 2: Overview of our method. Our approach begins with (a) pre-training the NeRF $\phi$ with a sampled inpainting prior $\tilde{\mathcal{I}}_p$ from Stable Diffusion $\theta$, detailed in Sec. 4.1. It then progresses to (b) latent-aligned inpainting $\tilde{\mathcal{I}}_i$ for multi-view images through Explicit Latents Alignment (ELA) and Implicit Latents Alignment (ILA), as described in Sec. 4.2. Finally, the NeRF is optimized using a patch-based hybrid loss strategy outlined in Sec. 4.3. Throughout the training process, we fix Stable Diffusion $\theta$ and update the scene-specific NeRF parameters $\phi$ only.

where $R_{\text{unmasked}}$ represent the unmasked pixels across all the training images. Then we sample a prior view $\mathcal{I}_p$ with its mask $\mathcal{M}_p$ and regularly inpaint it using Stable Diffusion $\theta$. For illustration, we replace of condition in Eq. 2 with two components used by Stable Inpainting Diffusion [71] as $e$ for the input prompt, and $\mathcal{I}'_p$ for the masked image that fed into the diffusion models. Hence the inpainting process can be formulated as:

$$z_p^{t-1} = z_p^t - \epsilon_\theta(z_p^t, t, \mathcal{I}'_p, e), \quad \text{for } t = T, \ldots, 1, \quad \text{with } z_p^T \sim \mathcal{N}(0, 1). \tag{4}$$

The inpainted image then can be obtained by $\tilde{\mathcal{I}}_p = \Omega(z_p^0)$. We then use a monocular depth estimator on $\tilde{\mathcal{I}}_p$ to get a depth map $\tilde{D}_p$. We regress the scale and offset parameters to align $\tilde{D}_p$ with the depth estimated from field $\phi$ on the unmasked pixels. Hence, we can introduce the geometry and appearance supervision of the inpainting prior $\tilde{\mathcal{I}}_p$ into the NeRF's optimization through:

$$\mathcal{L}_{\text{prior}}(\phi) = \sum_{r \in R_{\text{masked(p)}}} \left\| \hat{C}_\phi(r) - C(r) \right\|_2^2 + \left\| \hat{D}_\phi(r) - \tilde{D}_p(r) \right\|_2^2, \tag{5}$$

where $R_{\text{masked(p)}}$ denoted the masked (inpainted) pixels of $\tilde{\mathcal{I}}_p$, and $\hat{D}_\phi$ is the depth estimated by NeRF. This stage is depicted in Fig. 2(a).

## 4.2 Stage 2: Latents Alignment

In this section, we introduce our key approach to condition the additional inpainted frames to have an inpainting feature based on the prior $\tilde{\mathcal{I}}_p$. As discussed before, in deterministic sampling of the diffusion inpainting model $\theta$, the generation structure and layout highly depend on (1) initial **sampled** noise $z_i^T$ and (2) intermediate **predicted** latents $\{z_i^{t-1}\}_{t=1}^T$. Hence if we can align the latents from different views with the prior one, the model is likely to generate multi-view consistent latent $z_i^0$, hence image $\tilde{\mathcal{I}}_i$. In this section, we discuss how to align two terms respectively.

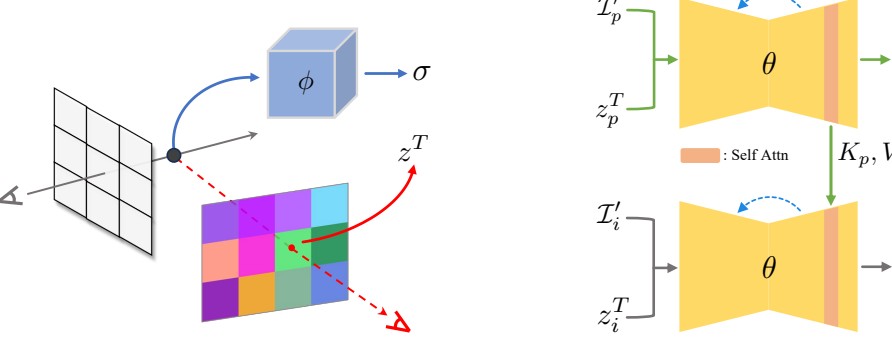

| (a) Explicit Initial Latent Alignment (ELA) | (b) Implicit Intermediate Latents Alignment (ILA) |

Figure 3: Illustration of two types of Latent Alignment. This figure depicts the Explicit Latents Alignment (ELA) and Implicit Latents Alignment (ILA) processes, as detailed in Sec. 4.2.

**Explicit Initial Latent Alignment (ELA)** Given the sampled latent of prior view $z_p^T$ used in Stage 1 (Sec. 4.1), we could leverage geometric information to explicitly align the initial latent in 3D space. Given that estimated depth $\tilde{D}_p$, one possible solution is to warp the $z_p^T$ into other views using camera matrices. However, $\tilde{D}_p$ is not guaranteed to be accurate hence such hard projection could yield significant errors. Alternatively, we propose to leverage the pre-trained NeRF $\phi$, as it's a naturally 3D-consistent representation. Specifically, to sample a resolution-grain initial latent $z^T(r)$, we utilize the original formulation of volume rendering (Eq.1) but with the substitution of color $c$ by latent $z^T$. We query the density $\sigma$ from $\phi$, and acquire $z^T$ by reprojecting the sampled point to the image plane of the prior latent view $z_p^T$:

$$z^T(r) = \sum \Gamma_i \Big( 1 - \exp \big( - \sigma(\tau_i)\delta(\tau_i) \big) \Big) z^T(\tau_i), \quad \text{with } z^T(\tau_i) = f_{p,i}(z_p^T, \tau_i), \tag{6}$$

where $f_{p,i}$ denote camera perspective projection according to $p$ and $i$ camera matrices. Such soft projection could avoid error accumulation in the inpainting process, and reduce the precision burden on the depth estimator. We illustrate this process in Fig. 3a. There are two key reasons why we propose fine-tuning the NeRF and using it as a geometric prior for ELA: (a) After finetuning the NeRF, the geometric is represented by NeRF as a sharp (low variance) unimodal distribution on the ray. Consequently, the aggregated feature remains sharp, preserving the variations in the initial latents. (b) We empirically found the depth prior inferred by the monocular depth estimator is not perfectly aligned with the NeRF. Fine-tuning the NeRF can also benefit this depth prior. Since NeRF learns relatively certain geometry in the known (unmasked) areas, this geometry constraint can improve the geometry of neighboring inpainted (masked) areas due to their geometric proximity. We compromise the view-dependent effect in NeRF within the ELA module. Due to the heuristic nature of diffusion models, incorporating such view-dependent effects into diffusion models' output remains elusive.

**Implicit Intermediate Latents Alignment (ILA)** While the initial latent could be aligned using the explicit method, intermediate latents are predicted by denoising network $\epsilon_\theta$ which is hard to control. We address this issue by exploring the conditioning mechanism of the denoising network in Stable Diffusion [71]. Recall that in Eq. 4, denoising network $\epsilon_\theta$ relies on the input prompt $e$ and masked image $\mathcal{I}_i'$ to predict the noise occurrence in the current step. While we can use the unified prompt for all views to align the text condition in cross-attention of $\epsilon_\theta$, the masked images $\{\mathcal{I}_i'\}_{i=1}^N$ are inherently different due to multi-view nature. Note that $\mathcal{I}_i'$ condition is introduced based on spatial self-attention (SA) [90] in the U-Net:

$$\text{SA}(Q_i, K_i, V_i) = \text{Softmax}\left(\frac{Q_i K_i^T}{\sqrt{d}}\right) V_i, \tag{7}$$

where $Q_i$ obtained from each spatial resolution of the latent, $K_i, V_i$ are derived from corresponding latent encoded from masked image $\mathcal{I}_i'$. We can impose the coherence of the denoising step by introducing cross-view attention (CVA) of the prior view (Fig. 3b):

$$\text{CVA}(Q_i, K_p, V_p) = \text{Softmax}\left(\frac{Q_i (K_p)^T}{\sqrt{d}}\right) V_p, \tag{8}$$

where $K_p, V_p$ are from masked base image $\mathcal{I}'_p$. We can then implicitly align the denoising step by replacing the original SA with a weighted sum from SA and CVA, i.e. $\lambda_a * \text{SA}(Q_i, K_i, V_i) + (1 - \lambda_a) * \text{CVA}(Q_i, K_p, V_p)$. Through this technique we ensure that the intermediate latents $\{z_i^{t-1}\}_{t=1}^T$ are also conditioned on the prior $\{z_p^{t-1}\}_{t=1}^T$, while retain its distinctiveness due to the individual viewpoint. The rationale of replacing $KV$ with "prior" $p$, but not $Q$ is that the appearance information $(V)$ of the prior image should be considered when inpainting the other views, with the amount of information propagation is weighted by its attention key value $(K)$. The attention query value comes from the current inpainting view $i$, $Q_i$, representing the information the current inpainting for view $i$ is searching for. Together with $K_p$, it decides how much attention the view $i$ inpainter should place on the prior view, and finally incorporates the information of the prior view $V_p$ into view $i$.

### 4.3 Stage 3: Joint Optimization

As the original intention of this work, we seek to distill the inpainted views into NeRF $\phi$ in a way such that the high-fidelity is preserved as much as possible as the unmasked region. While some priors work in 3D editing [25, 96, 97, 93] propose to update the training set iteratively until converge, we empirically find it not suitable for inpainting task since the loss of fidelity is significant and could fall into local optima. Hence we propose to inpaint a subset of training images at once and regard them as supplementary guidance using a patch-based hybrid loss:

$$\mathcal{L}_{\text{patch}}(\phi) = \sum_{\rho \in \mathcal{P}_{\text{sub}}} \left\| \hat{I}_\phi(\rho) - \tilde{\mathcal{I}}(\rho) \right\|_1 + \mathcal{L}_{\text{lpips}}(\hat{I}_\phi(\rho), \tilde{\mathcal{I}}(\rho)) + \mathcal{L}_{\text{adv}}(\hat{I}_\phi(\rho), \tilde{\mathcal{I}}(\rho)), \tag{9}$$

where $\rho$ is a patch sample from the masked area of subset views $\mathcal{P}_{\text{sub}}$, $\hat{I}_\phi(\rho)$ is NeRF predicted patch, and $\mathcal{L}_{\text{lpips}}, \mathcal{L}_{\text{adv}}$ are perceptual distance LPIPS [106] and adversarial loss [24]. Here LPIPS is utilized to address geometry mismatches, while adversarial loss is employed to preserve high-frequency details. As shown in Fig. 2, the final optimization objective is:

$$\mathcal{L}(\phi) = \mathcal{L}_{\text{rec}}(\phi) + \mathcal{L}_{\text{prior}}(\phi) + \lambda_{\text{patch}} \mathcal{L}_{\text{patch}}(\phi). \tag{10}$$

The patches are uniformly sampled within the bounding box of the mask, with a size of 256×256. Therefore, only the inpainted area is being optimized by the patch loss.

## 5 Experiment

### 5.1 Evaluation Setting

**Dataset:** Aligning with methodologies employed in prior works [53, 51, 109], our experiments utilize the SPIn-NeRF dataset [53], selected for its comprehensive ground truth availability. This dataset is specifically designed for object removal evaluations and comprises 10 scenes. Each scene includes 60 images featuring an unwanted object (training views) and 40 images from which the object has been removed (test views). For both the training and test views, human-annotated masks indicating the object region are available. We further collected 9 forward-facing scenes with manually annotated masks to evaluate the effectiveness of our method. This dataset includes 4 indoor scenes and 5 outdoor scenes. In the training set, the masked region contains the unwanted object, while the test set contains the ground truth background in the masked region.

**Baselines:** In our study, we benchmark our method against a variety of established 3D inpainting approaches to ascertain its relative performance. These include the perceptual-based SPIn-NeRF [53], tiling-based NeRFiller [93] (both multi-view guidance) and InFusion [109] (single-view guidance). To ensure a fair comparison, we employ the same inpainting diffusion models across all methods and maintain consistency in the number of denoising steps and used prompts. We utilized the source code provided by the authors and ran all the methods using one NVIDIA A100 (80G) GPU.

**Metrics:** To quantitatively evaluate the effectiveness of our approach, we employ two similarity metrics: LPIPS [106] and FID [28]. Additionally, we use MUSIQ [36], a sharpness metric that quantifies the clarity and detail retention in the edited images. Following established protocols from previous studies [53], all metrics are calculated specifically within the bounding boxes defined by the masks, focusing the evaluation precisely on the regions most affected by the object removal task.

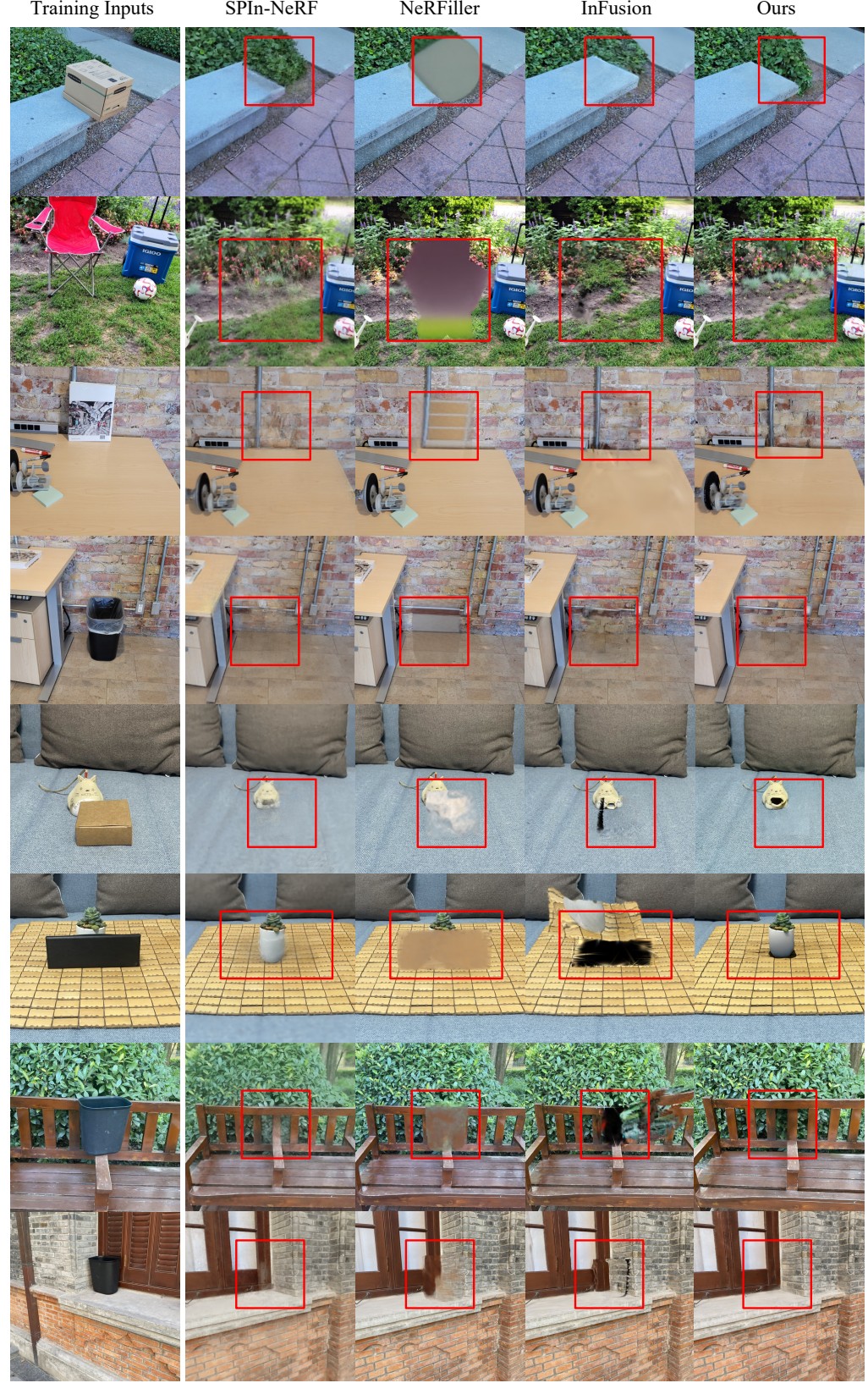

Training Inputs     SPIn-NeRF     NeRFiller     InFusion     Ours

Figure 4: Qualitative results on the SPIn-NeRF the self-collected dataset.

## 5.2 Main Results

We first present the quantitative results in Tab. 1, where our method outperforms all baselines in terms of similarity metrics and sharpness. Our approach also excels in qualitative assessments, as demonstrated in Fig. 4. It is important to note that in simpler scenes with low variability in inpainting results, where the inconsistency issue is less pronounced (first row), most methods perform adequately. In other cases, high-frequency loss is observed in multi-view-based methods (SPIn-NeRF and NeRFiller). NeRFiller [93], through its use of multiple joint denoising steps, ensures consistency but often produces overly smooth outputs that lack fine details. It is noteworthy that the single-view-based method, InFusion [109], relies on one view and its depth to represent the entire scene. It performs well when geometry estimation is accurate. However, its performance deteriorates in scenarios where depth accuracy is compromised, leading to geometry artifacts (sixth and seventh rows). This underscores the critical role of multi-view supervision in addressing such challenges. By incorporating consistent multi-view supervision, our method remains effective even when depth or geometry is inaccurate, achieving robust and promising results. This explains why our method shows little difference from InFusion when the geometry is accurate (first and third row), but excels when the depth is inaccurate. Additionally, the exclusive reliance on perceptual loss by SPIn-NeRF [53] fails to fully address the multi-view inconsistencies introduced by inpainting diffusion models, often resulting in a blurred effect, particularly visible in the third and fourth rows. To further validate our findings, we conducted a user study based on the SPIn-NeRF dataset, focusing on the coherence of the background within the inpainted area, the fidelity of detail preservation in the inpainted region, and overall preference. The results of this study are summarized in Tab. 2. This evaluation clearly demonstrates superior performance across all assessed criteria.


<div>

Table 1: Quantitative Results

| Method | LPIPS ↓ | FID ↓ | MUSIQ ↑ |
|---|---|---|---|
| SPIn-NeRF [53] | 0.54 | 185.63 | 38.69 |
| NeRFiller [93] | 0.71 | 315.83 | 32.60 |
| InFusion [109] | 0.62 | 153.77 | 39.29 |
| Ours | **0.49** | **130.92** | **50.97** |

</div>
<div>

Table 2: User Study

| Method | Coherence | Fidelity | Overall |
|---|---|---|---|
| SPIn-NeRF [53] | 22.72% | 20.45% | 21.82% |
| NeRFiller [93] | 2.73% | 4.33% | 2.50% |
| InFusion [109] | 27.50% | 24.77% | 25.00% |
| Ours | **47.05%** | **50.45%** | **50.68%** |

</div>
</div>

## 5.3 Ablation Studies

We initially demonstrate the efficacy of our latents alignment approach with an example in Fig. 5. The first column displays the inpainting prior (sampled view), and the subsequent columns show the same training image being inpainted under different conditions. Notably, the variant without ELA (w/o ELA) retains colors similar to the prior but fails to preserve the texture structure. Conversely, the version without ILA (w/o ILA) maintains structural integrity but lacks appearance consistency with the prior. Our method effectively merges the strengths of both mechanisms, resulting in inpaintings that are highly consistent and cohesive across all evaluated aspects.

| Inpainting prior | Ours w/o ELA | Ours w/o ILA | Ours |
|---|---|---|---|

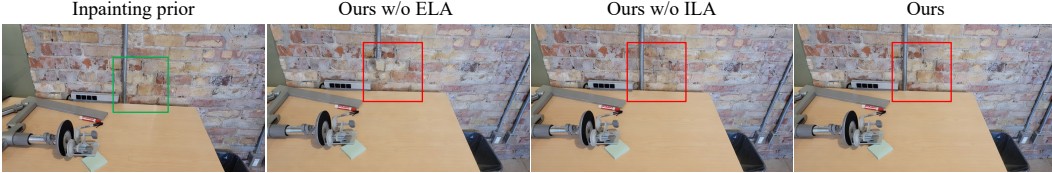

Figure 5: Ablation study on latent aligned onpainting. 2D Inpainting results when key components of our proposed method are omitted. Naive inpainting using Stable Diffusion can refer to Fig. 1.

We conducted further ablation studies to underscore the importance of our key design elements in object removal tasks. The quantitative and qualitative results, showcased in Tab. 3 and Fig. 6, clearly indicate the impact of each component. Notably, removing ELA leads to geometry mismatches in the NeRF outputs (w/o ELA), while deactivating ILA results in blurry coloration (w/o ILA). This observation confirms our initial findings: the initial latents primarily influence the inpainting's structural pattern, whereas the intermediate denoising steps largely affect its appearance, including

colour nuances. Additionally, our patch-based loss plays a crucial role in the optimization process (w/o $\mathcal{L}_{\text{patch}}$). Specifically, the $\mathcal{L}_{\text{lpips}}$ loss helps to alleviate geometry mismatches (w/o $\mathcal{L}_{\text{lpips}}$), and the $\mathcal{L}_{\text{adv}}$ serves as a detail-preserving supervisor (w/o $\mathcal{L}_{\text{adv}}$). These results highlight the effectiveness of our design choices in enhancing the overall quality and coherence of the inpainted outputs.

Table 3: Quantitative Results of Ablation Study.

| Method | LPIPS ↓ | FID ↓ | MUSIQ ↑ |
|---|---|---|---|
| Ours w/o ELA | 0.52 | 133.09 | 48.90 |
| Ours w/o ILA | 0.50 | 141.78 | 49.70 |
| Ours w/o $\mathcal{L}_{\text{patch}}$ | 0.73 | 293.32 | 33.76 |
| Ours w/o $\mathcal{L}_{\text{lpips}}$ | 0.55 | 223.31 | 46.07 |
| Ours w/o $\mathcal{L}_{\text{adv}}$ | 0.51 | 134.70 | 49.88 |
| Ours full model | **0.49** | **130.92** | **50.97** |

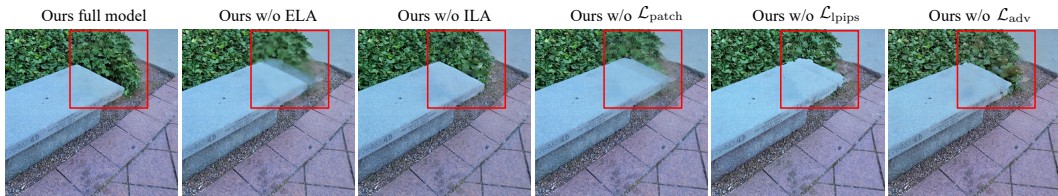

Figure 6: Ablation study on design choices based on rendering quality. This figure displays rendering results from NeRF when key components are individually removed from our full model.

## 6  Conclusion

In this work, we demonstrate the significant improvement achieved through our novel latents alignment approach in 3D object removal. By integrating both explicit and implicit latent alignment mechanisms, we have successfully addressed key challenges associated with geometry mismatches and color inconsistencies that are prevalent in the baselines, enhancing the fidelity and detail of the inpainted 3D scenes. The improvements achieved through our work offer significant societal benefits, such as enhanced editability of radiance fields. However, it also poses risks, including the potential perpetuation of biases and discrimination. If the data used to train diffusion models is biased, our approach could inadvertently reinforce these biases.

Despite notable advancements, our method has limitations: (1) It struggles with full 3D consistency, especially on high-frequency details, due to the constraints of applying 2D diffusion models to multi-view data. Future work could address this by integrating multi-view training into 2D inpainting diffusion models or leveraging true 3D generative models. (2) It is tailored for forward-facing scenes, limiting its applicability to diverse 360° views. Further exploration of latent relationships for broader view coverage is needed. (3) Predefined masks are currently required. Integrating advanced 3D perception methods [66, 67] could enhance accuracy and flexibility, enabling precise language-driven interactions and creating a more automated, user-friendly framework for neural 3D scene editing.

## 7  Acknowledgements

This research was mainly undertaken using the LIEF HPC-GPGPU Facility hosted at the University of Melbourne. This Facility was established with the assistance of LIEF Grant LE170100200. This research was also partially supported by the Research Computing Services NCI Access scheme at the University of Melbourne. DH was supported by the Melbourne Research Scholarship from the University of Melbourne. FL is supported by the Australian Research Council (ARC) with grant numbers DP230101540 and DE240101089, and the NSF&CSIRO Responsible AI program with grant number 2303037.

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

# A Supplemental Material for "In-N-Out: Lifting 2D Diffusion Prior for 3D Object Removal via Tuning-Free Latents Alignment"

## A.1 Implementation Detail

For the inpainting network, we employ the stable-diffusion-2-inpainting model [71], which encodes a masked image into the same dimensional latent space and integrates conditions via concatenation. We set the denoising steps for inpainting at 20. To achieve better generalization, we propose sampling the base frame according to the geometrical centroid of the training camera poses, meaning the camera that sits most centrally among the training views. However, we found that Stable Diffusion occasionally inpaints strange artifacts in the masked region. To mitigate this, we propose sampling $n$ candidate views around the geometrical centroid and selecting the one with the highest similarity votes. This approach automatically avoids such occasion artifacts without human intervention. In our implementation, we used five candidate views, and the similarity was calculated using perceptual hashing. In the reprojection procedure of ELA, we adjust the camera intrinsics to match the latent dimensionality. Furthermore, to refine the ILA mechanism, we incorporate Cross-View Attention (CVA) into every self-attention layer of the inpainting model. Each step in this modified approach is controlled with $\lambda_a$ set at 0.2.

For our 3D representation (NeRF) implementation, we utilize the "nerfacto" framework proposed by NerfStudio [83]. To ensure stable training, we deactivated the view-dependent effect. We pre-train the NeRF using 10000 iterations in stage 1 and jointly optimize it using 5000 iterations in stage 3. Our monocular depth estimation adopts DepthAnything [98], complemented by the depth loss outlined in DS-NeRF [17]. Moreover, we employ StyleGAN2 discriminator [35] to implement adversarial loss.

## A.2 Sensitivity Analysis

We conducted several sensitivity analyses regarding the base view selection, $\lambda_a$ in ILA, and the subset selection. Due to the computational burden, we conduct the sensitivity analysis on six out of ten scenes with higher inpainting variability from the SPIn-NeRF dataset.

**(a) Base View Selection:**

To achieve better generalization, we propose sampling the base frame according to the geometrical centroid of the training camera poses, meaning the camera that sits most centrally among the training views. However, we found that Stable Diffusion occasionally inpaints strange artifacts in the masked region. To mitigate this, we propose sampling $n$ candidate views around the geometrical centroid and selecting the one with the highest similarity votes. This approach automatically avoids such occasion artifacts without human intervention. In our implementation, we used five candidate views, and the similarity was calculated using perceptual hashing.

We tested our results under different settings (candidate numbers): 3, 5, 7, and 9. The base frame selection algorithm proved to be robust, with our algorithm typically yielding the same base frame. However, another factor influencing this step is the random seed. Setting different seeds causes the 2D inpainting model to produce different results, leading to different base frames being selected. We tested our methods under five different seeds, and the final scores are reported in Table 4. While different seeds cause the final NeRF to differ in the appearance of the masked region, the consistency of the multi-view inpainting results remains robust, resulting in minimal variance in the evaluation scores.

**(b) $\lambda_a$ in ILA:**

To effectively examine the effect of the hyper-parameter $\lambda_a$ in ILA, we evaluated our method's rendering quality with different $\lambda_a$ values of 0.2, 0.4, 0.6, and 0.8. The metrics are reported in Table 5. Quantitatively, the results are consistent across different $\lambda_a$ values, indicating that the effect of this hyper-parameter is relatively small. This conclusion is also supported by qualitative results. Larger $\lambda_a$ values tend to produce slight variations in some small regions, but the global structure and semantics are preserved. This stability is attributed to the significant role of the initial latent alignment in ELA, which effectively aligns the underlying inpainting structure, thereby maintaining low variability in appearance. Additionally, the self-attention layer, where cross-view attention is introduced, does not dominate the entire Stable Diffusion Unet. It is balanced by the presence of

Table 4: Sensitivity analysis on the prior inpainting results and prior view selection. Results are evaluated on the SPIn-NeRF dataset with different random seeds.

| Seed | LPIPS ↓ | MUSIQ ↑ | FID ↓ |
|------|---------|---------|-------|
| 1 | 0.46 | 46.61 | 264.91 |
| 2 | 0.44 | 48.04 | 255.29 |
| 3 | 0.44 | 46.47 | 262.09 |
| 4 | 0.44 | 45.72 | 261.04 |
| 5 | 0.46 | 48.65 | 258.50 |
| Avg | 0.45 | 47.10 | 260.37 |
| Std | 0.01 | 1.21 | 3.657 |

other (residual and linear) layers, ensuring cross-view attention does not override the signal during the denoising process. Hence we simply set $\lambda_a$ as 0.2 in our implementation.

Table 5: Sensitivity analysis on $\lambda_a$ used in ILA.

| $\lambda_a$ | LPIPS ↓ | MUSIQ ↑ | FID ↓ |
|-------------|---------|---------|-------|
| 0.2 | 0.44 | **47.11** | **261.62** |
| 0.4 | **0.44** | 46.76 | 264.91 |
| 0.6 | 0.44 | 46.47 | 264.37 |
| 0.8 | 0.45 | 46.33 | 265.10 |
| Avg | 0.44 | 46.67 | 264.00 |
| Std | 0.01 | 0.35 | 1.62 |

**(c) Subset Selection:**

We found that for reconstruction tasks, more views can enhance quality; however, for generation tasks, using the entire set of images can introduce unnecessary inconsistencies. Therefore, we propose selecting the subset according to the distribution of camera viewpoints.

We evenly split the viewpoints into 12 groups based on the base view's camera space (evenly 2 on the x and y axes and 3 on the z-axis) and select 50 percent within each group according to perceptual hashing similarity to the base view. This approach avoids redundant views introducing supervision conflicts while covering different viewpoints for effective supervision.

We also evaluated our method based on different percentages, as reported in Table 6. The quantitative scores are quite close, indicating that for most scenes, the difference isn't significant. For one complex scene with extremely high frequencies, setting the percentage too low (0.2) yields artifacts in the test view due to insufficient viewpoint coverage. Conversely, setting the percentage too high (0.8) introduces appearance conflicts due to the high variability of the inpainted results.

Overall, for most scenes, the subset selection algorithm is robust due to the consideration of viewpoints distribution. For extreme cases, careful selection of the percentage might be necessary. However, values between 0.5 and 0.7 remain a reliable choice.

Table 6: Sensitivity analysis on proportion of images selected for the subset.

| Percentage | LPIPS ↓ | MUSIQ ↑ | FID ↓ |
|------------|---------|---------|-------|
| 0.2 | 0.46 | 45.98 | 265.48 |
| 0.4 | 0.44 | 46.32 | 264.91 |
| 0.6 | **0.44** | **47.11** | **261.62** |
| 0.8 | 0.45 | 46.47 | 263.20 |

**(d) $\lambda_{patch}$ in patch loss:**

To assess the sensitivity of the patch loss multiplier $\lambda_{patch}$, we evaluated the method's performance using various values of $\lambda_{patch}$: 0.001, 0.005, 0.01, 0.05, and 0.1. The results are reported in Table 7. Analysis of the table indicates that varying $\lambda_{patch}$ leads to similar performance across different

settings, with a low standard deviation of the metrics. However, there is an observable trend where setting $\lambda_{patch}$ too low or too high adversely affects performance. The multiplier $\lambda_{patch}$ is critical as it determines the extent of influence multi-view images have on the NeRF. Insufficient multi-view supervision can lead to inadequate training, whereas excessive supervision may result in conflicting inputs. Consequently, we have set $\lambda_{patch}$ at 0.01 in our implementation for optimal balance.

Table 7: Sensitivity analysis on $\lambda_{patch}$ used for patch loss.

| $\lambda_{patch}$ | LPIPS $\downarrow$ | MUSIQ $\uparrow$ | FID $\downarrow$ |
|---|---|---|---|
| 0.001 | 0.46 | 46.078 | 263.32 |
| 0.005 | 0.45 | 47.08 | 262.43 |
| 0.010 | **0.44** | **47.11** | **261.62** |
| 0.050 | 0.47 | 44.93 | 265.31 |
| 0.100 | 0.49 | 44.05 | 277.36 |
| Avg | 0.46 | 45.85 | 266.01 |
| Std | 0.02 | 1.35 | 6.49 |

## A.3 More Qualitative Results

This section presents extended qualitative results from our experiments on the SPIn-NeRF Dataset. Fig. 7 and Fig. 8 showcase a series of multi-view comparative inpaintings.

## A.4 Details on User Study and Impact

To comprehensively evaluate our method using human subjects, we conducted a user study focusing on three aspects: (1) Background Coherence — assessing whether the inpainted area blends seamlessly with the remaining background, (2) Detail Preservation — determining if the inpainted area retains high-fidelity details, and (3) Overall Quality — gauging participants' preference rates for the inpainted results. For each method, we presented users with two multi-view test images from each scene and instructed them to choose the method that best met the criteria for each aspect. Clear instructions were provided to ensure participants understood the rating process. An example screenshot of the study interface is shown in Fig. 9.

The user study we conducted focused solely on collecting participants' preferences regarding different inpainting results, involving no sensitive or personal data collection beyond their aesthetic judgments. The study's design was inherently low-risk as it required participants to simply view and evaluate digital images based on their visual appeal and perceived quality. Furthermore, the participation was entirely voluntary, with clear instructions provided, allowing participants to withdraw at any time without any consequence. Given these factors, the potential for harm or discomfort to participants was negligible, ensuring the study maintained a minimal risk profile.

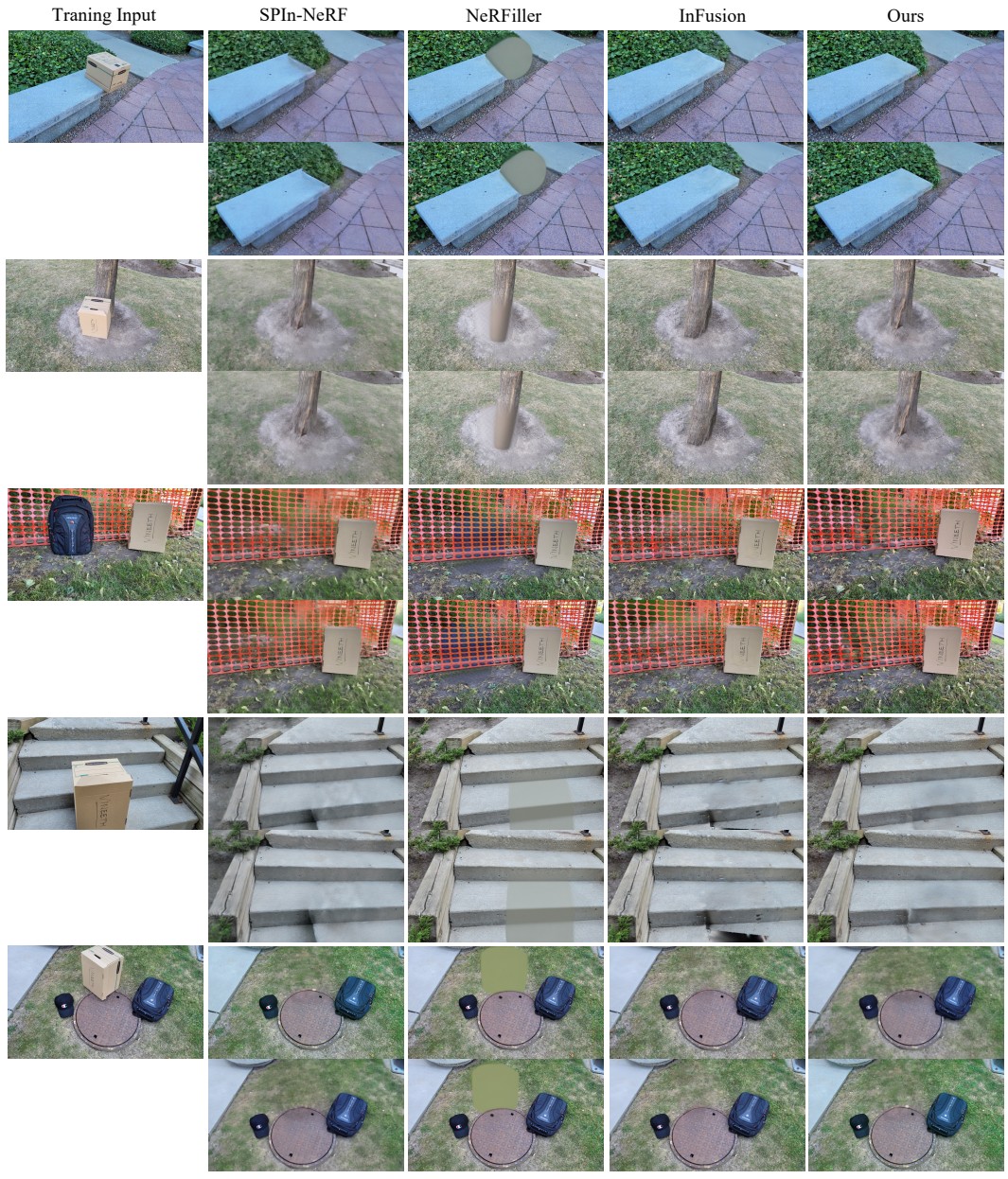

Figure 7: Additional Qualitative Results on the SPIn-NeRF Dataset.

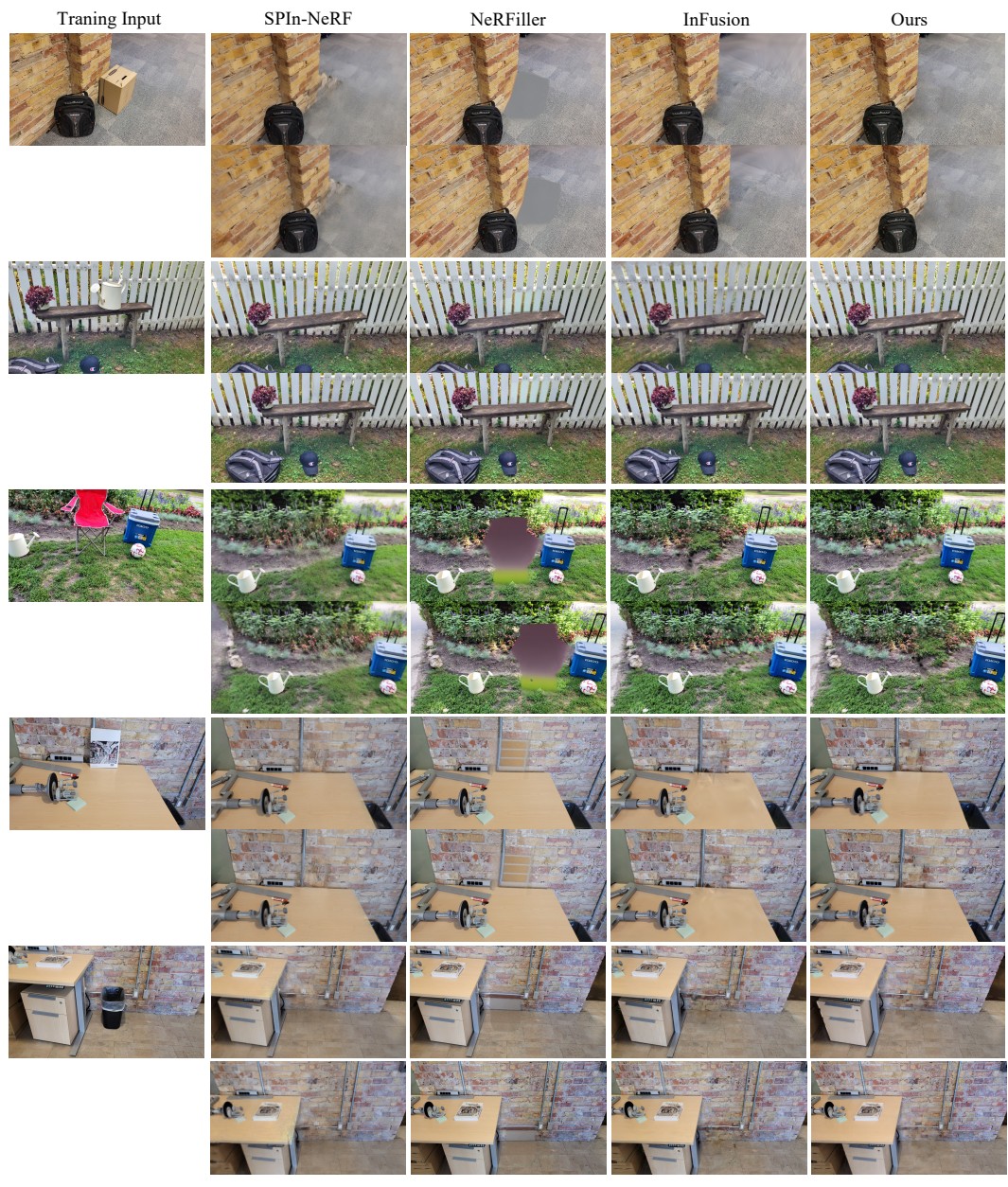

Figure 8: Multi-view Qualitative Results on the SPIn-NeRF Dataset.

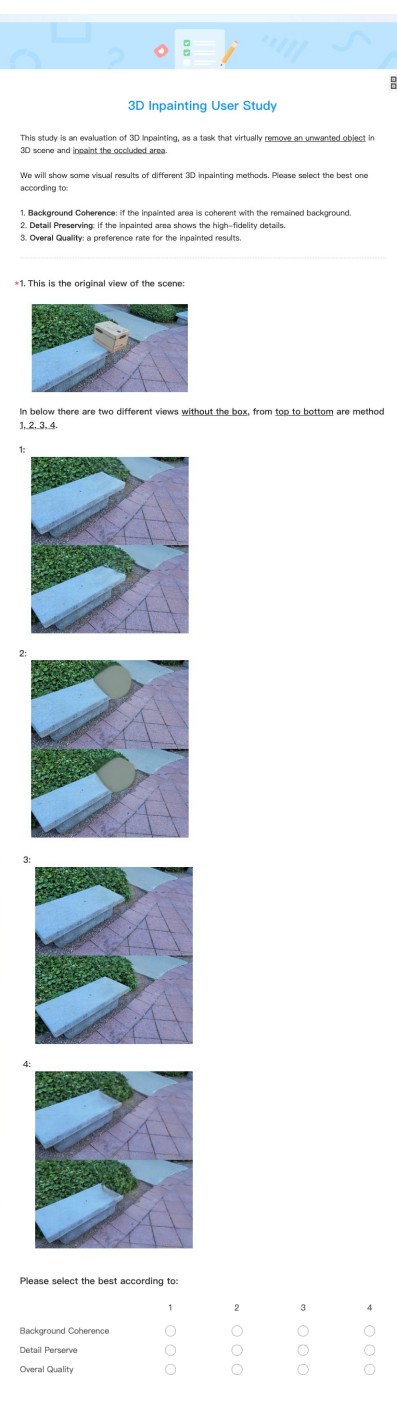

Figure 9: Example of User Study.

