# OpenReview forum: "In-N-Out: Lifting 2D Diffusion Prior for 3D Object Removal via Tuning-Free Latents Alignment"
_NeurIPS.cc/2024/Conference — NeurIPS 2024 poster_

### Official Review · Reviewer_P7tn · 2024-07-07

**Soundness:** 3
**Presentation:** 1
**Contribution:** 3
**Rating:** 6
**Confidence:** 5

**Summary:**

The paper introduces a novel approach termed "In-N-Out" for enhancing the performance of 3D object removal tasks by leveraging tuning-free latents alignment. The authors have addressed the challenges of geometric mismatches and color inconsistencies prevalent in existing methods.

**Strengths:**

The authors have conducted an extensive review of related work and clearly delineated the distinctions and connections between their work and existing literature.The explicit and implicit latents alignment proposed by the authors is an intriguing concept that could inspire work in related domains.

**Weaknesses:**

The paper exhibits signs of being finished in a rush, which affects the overall quality of academic writing, and is hard to follow.
For example，
1. Line172，Omega(·) is not defined
2. Formula (5), D_hat is note defined
3. Line 208， by replace -> by replacing
The paper might has the potential to contribute to the field but requires significant revisions to improve the academic writing quality.

The method involves several hyperparamete. The paper does not discuss the sensitivity analysis or the robustness of the method against variations in hyperparameters.  Especially，the lambda_a, which plays an essential role in ILA.

**Questions:**

1. In ELA module, using latent z^T to replace color c for volume rendering. Will this compromise the view-dependent in NeRF? In NeRF，the color  is dependent on the position x and direction d, whereas in ELA, the z^T is merely reprojected to the image plane.

2. How do you choose lambda_a in ILA.

**Limitations:**

The method involves several hyperparamete. The paper does not discuss the sensitivity analysis or the robustness of the method against variations in hyperparameters.

---

> ### Author Rebuttal · Authors · 2024-08-07
>
> - **1. Quality of academic writing.**
>
> Thanks for pointing out these typos. In line 172, Omega(·) is a decoder which is introduced in line 151, and  $\hat{D}_{\phi}$ is the depth estimated by NeRF. We hope these typos do not affect the clarity of the ideas presented in this paper.
>
> - **2. Compromise of view-dependent effect in ELA.**
>
> Yes, we compromise the view-dependent effect in NeRF within the ELA module, using NeRF only as a geometry prior to propagate and aggregate initial latents. Due to the heuristic nature of diffusion models, incorporating view-dependent effects into their output remains elusive.
>
> - **3. Lack of sensitivity analysis.**
>
> Thanks for the constructive feedback. We've conducted 4 sensitivity analysis regarding the base view selection, $\lambda_a$ in ILA, subset selection and $\lambda_\text{patch}$ for patch loss. Due to the computational burden, we conduct the sensitivity analysis on six out of ten scenes with higher inpainting variability from SPIn-NeRF dataset.
>
> ### (a) Base View Selection:
> To achieve better generalization, we propose to sample $n$ candidate views around the geometry centroid of the camera viewpoints and select the one with the highest similarity votes, automatically avoiding artifacts without human intervention. We used 5 candidate views, with similarity calculated using perceptual hashing.
>
> We tested our results under different settings (candidate numbers): 3, 5, 7, and 9. The selection algorithm proved to be robust, with our algorithm typically yielding the same base frame. However, another factor influencing this step is the random seed. Setting different seeds causes the 2D inpainting model to produce different results, leading to different base frames being selected. We tested our method under five different seeds, with final scores reported in the table below and additional qualitative results in Figure 2 of the rebuttal PDF. While different seeds affect the appearance of the masked region in the final NeRF, the consistency of multi-view inpainting remains robust, resulting in minimal variance in evaluation scores. We will explicitly clarify this point in the revision.
>
> >|Seed|LPIPS↓|MUSIQ↑|FID↓|
> >|------|---------|---------|-------|
> >| 1|0.46|46.61|264.91|
> >| 2|0.44|48.04|255.29|
> >| 3|0.44|46.47|262.09|
> >| 4|0.44|45.72|261.04|
> >| 5|0.46|48.65|258.50|
> >|*Avg*|*0.45*|*47.10*|*260.37*|
> >|*Std*|*0.01*|*1.21*|*3.657*|
>
> ### (b) $\lambda_a$ in ILA:
>
> To examine the effect of the hyper-parameter $\lambda_a$ in ILA, we evaluated our method's rendering quality with varies $\lambda_a$ value. The metrics are reported in below table. The results are consistent across different $\lambda_a$ values, indicating a relatively small effect. This conclusion is supported by qualitative results in Figure 3 of the rebuttal PDF, where larger $\lambda_a$ values produce slight variations in small regions, but the global structure and semantics are preserved.
>
> This stability is attributed to the significant role of the initial latent alignment in ELA, which effectively aligns the underlying inpainting structure, thereby maintaining low variability in appearance.  Additionally, the self-attention layer, where cross-view attention is introduced, does not dominate the entire Stable Diffusion Unet. It is balanced by the presence of other (residual and linear) layers, ensuring cross-view attention does not override the signal during the denoising process. Hence we simply set $\lambda_a$ in our implementation. We will explicitly clarify this point in the revision.
>
> >|$\lambda_a$|LPIPS↓|MUSIQ↑|FID↓|
> >|-------------|---------|---------|---------|
> >|0.2|0.44|**47.11**|**261.62**|
> >|0.4|**0.44**|46.76|264.91|
> >|0.6|0.44|46.47|264.37|
> >|0.8|0.45|46.33|265.10|
> >|*Avg*|*0.44*|*46.67*|*264.00*|
> >|*Std* |*0.01*|*0.35 *|*1.62*|
>
> ### (c) Subset Selection:
>
> We propose selecting a subset for stage 3 training based on the distribution of camera viewpoints. We evenly split the viewpoints into 12 groups based on the base view's camera space (evenly 2 on the x and y axes and 3 on the z axis) and select 50% views within each group according to perceptual hashing similarity to the base view. This approach avoids redundant views introducing supervision conflicts, while covering different viewpoints effectively.
>
> We evaluated our method with varies selection percentages, as reported in the table below. The scores are close, indicating minimal differences for most scenes. For one complex scene with high frequencies, setting the percentage too low (0.2) yields artifacts due to insufficient viewpoint coverage, while setting it too high (0.8) introduces appearance conflicts due to variability in inpainted results. These results are visualized in Figure 4 of the rebuttal PDF.
>
> >|Percentage|LPIPS↓|MUSIQ↑|FID↓|
> >|------------|---------|---------|---------|
> >|0.2|0.46|45.98|265.48|
> >|0.4|*0.44*|46.32|264.9|
> >|0.6|**0.44**|**47.11** |**261.62**|
> >|0.8|0.45|*46.47*|*263.20*|
> >|*Avg*|0.44|46.67|264.00|
> >|*Std*|0.01|0.35|1.62|
>
> ### (d) $\lambda_{patch}$ in patch loss:
>
> To assess the sensitivity of the patch loss multiplier $\lambda_{patch}$ in Eq. 10 of the main paper, we evaluated the method's performance using varies $\lambda_{patch}$ values. The results in below table show similar performance across different settings, with low standard deviation. However, setting $\lambda_{patch}$ too low or too high adversely affects performance. $\lambda_{patch}$ is critical as it influences the extent of multi-view supervision on NeRF; insufficient supervision leads to inadequate training, while excessive supervision causes conflicts. We set $\lambda_{patch}$ at 0.01 in our implementation for optimal balance.
>
> >|$\lambda_{patch}$|LPIPS↓|MUSIQ↑|FID↓|
> >|-------------------|---------|---------|---------|
> >|0.001|0.46|46.07|263.32|
> >|0.005|0.45|47.08|262.43|
> >|0.01|**0.44**|**47.11**|**261.62**|
> >|0.05|0.47| 4.93|265.31|
> >|0.1|0.49|44.05|277.36|
> >|*Avg*|*0.46*|*45.85*|*266.01*|
> >|*Std*|*0.02*|*1.35*|*6.49*|

---

> > ### Comment · Reviewer_P7tn · 2024-08-11
> >
> > The author has conducted additional experiments, and the results now appear satisfactory. Therefore, I have decided to increase my score by one point.

---

> > > ### Author Response · Authors · 2024-08-11
> > >
> > > Thank you for reconsidering the revised experiments and for acknowledging the improvements in the results. We appreciate your decision to increase the score and are grateful for your thorough feedback.

---

> > > > ### Comment · Reviewer_P7tn · 2024-08-13
> > > > **Future work discussion.**
> > > >
> > > > The current model uses multi-view masks as input. In future work, it could be combined with 3D perception methods, such as LangSplat [1] and GOI [2], which are two SOTA open-vocabulary 3D  perception works. I hope you can include a discussion in the camera-ready version about the potential of integrating 3D perception approaches.
> > > >
> > > > [1] Langsplat: 3d language gaussian splatting, CVPR 2024
> > > > [2] GOI: Find 3D Gaussians of Interest with an Optimizable Open-vocabulary Semantic-space Hyperplane, ACM MM2024

---

> > > > > ### Author Response · Authors · 2024-08-13
> > > > > **Future work discussion.**
> > > > >
> > > > > Thank you for reviewer's valuable insights. We completely agree that 3D perception and segmentation methods are highly related to our task. Indeed, accurate 3D perception/segmentation methods are a prerequisite step, as the quality of multi-view masks directly impacts the performance of inpainting outcomes. In practice, we find these methods play a crucial role in machine vision automation and are applicable to a variety of downstream tasks.
> > > > >
> > > > > Integrating 3D perception approaches such as LangSplat[1] and GOI[2] with object removal and 3D inpainting tasks can significantly enhance the effectiveness and flexibility of these applications. These advanced perception methods, which efficiently map language queries to 3D spatial contexts, crucial for specifying and identifying the exact areas for removal or inpainting in complex scenes. For instance, using language-driven approach, a system can precisely locate and manipulate objects as per user instructions, enabling more intuitive and interactive design adjustments. Moreover, the open-vocabulary capabilities of these methods ensure that the system can adapt to a wide range of objects and scenes, potentially improving the automation and generalization of 3D inpainting tasks. In the future work, we could combine our work with 3D perception methods to enable a unified framework for 3D scene editting. This will largely enhance flexibility and user-friendliness. More details will be discussed in the revised version.

---

> > > > > > ### Comment · Reviewer_P7tn · 2024-08-13
> > > > > > **Decision**
> > > > > >
> > > > > > I appreciate the effectiveness and thoroughness of this work's experiments. After reading the author's rebuttal letter, which addressed most of my concerns and  promised to discuss the relationship with 3D perception works in the next revision, I have decided to raise my review score to 6—weak accept

---

> > > > > > > ### Author Response · Authors · 2024-08-13
> > > > > > >
> > > > > > > Thank you very much for your feedback and for reconsidering your review score. We are glad to hear that our rebuttal addressed most of your concerns. We are committed to further refining our discussion on the relationship with 3D perception works in the revision, and we appreciate your support and constructive comments as they greatly enhance our work's development.

---

### Official Review · Reviewer_f6e3 · 2024-07-11

**Soundness:** 3
**Presentation:** 3
**Contribution:** 3
**Rating:** 7
**Confidence:** 3

**Summary:**

The paper deals with 3D inpainting problem in the NeRF setup with 2D diffusion models. To achieve multiview consistency, they took a "inpaint-outstretch" strategy. They first inpaint one key frame, and conditioned on which, generate a view-consistent inpainted image set. Finally, they use the multiview images to train the final NeRF model, together with the reconstruction losses. The method is compared to a few recent works (Spin-NeRF, NeRFiller, InFusion) and achieved better results both qualitatively and quantitatively.

**Strengths:**

- The method is interesting and seems effective. They improve the consistency of multiview generation in two ways. The first is explicit latent alignment (ELA) that initializes the latent from the inpainted base frame. The second is ILA that post-hoc modifies the attention layer of stable diffusion models by replacing the key and value with the ones from the inpainted base frame. ILA does not need finetuning.

- The method is evaluated on the SPIn-NeRF dataset and the performance gain over existing methods is quite large, both quantitatively and via user study.

- There is also an ablation study that shows the importance of ELA, ILA, and the patch-based losses applied on the generated image sets.

- The method is well written and easy to follow.

**Weaknesses:**

- My biggest confusion is on the explanation of ILA.
  - I don't fully understand why replacing the KV with the base frame KV would encourage consistency. I would expect the style to be consistent but not the local geometry, where the latter is more important for 3d reconstruction.
  - What is the rationale of replacing KV with "prior" p, but not Q?
- The algorithm relies on a pre-defined base frame p. How is the base frame chosen, and how sensitive is the algorithm to the chosen frame?

**Questions:**

- Is Eq (10) applied to Stage 3? If so, what is the purpose of the prior loss L_prior? Once the inpainted base image is used to generate a multiview training set, I would expect this information to be unnecessary? Can this term be merged into L_patch?
- In L216, what is the subset? Why not inpaint the whole set of images and how much does it affect the results?

**Limitations:**

Yes

---

> ### Author Rebuttal · Authors · 2024-08-07
>
> - **1. Explanation of ILA**
>
> (a) The reviewer's intuition of ILA is entirely correct. ILA primarily contributes to appearance (color) consistency. Geometry consistency is achieved by ELA (the alignment of initial noise) since this element serves as a foundational structure (semantic) for the inpainting. This is verified in our ablation study in Figure 6 of the main paper. Removing ELA results in geometry mismatch (blurry boundaries), while dropping ILA achieves geometry alignment but causes color mismatch issues (blurry leaves). This is why we propose ELA and ILA together to ensure both geometry and appearance consistency.
>
> (b) The rationale of replacing $KV$ with "prior" $p$, but not $Q$ is that the appearance information (**$V$**) of the prior image should be considered when inpainting the other views, with amount of information propagation is weighted by its attention key value (**$K$**). The attention query value comes from the current inpainting view $i$, $Q_i$ , representing the information the current inpainting for view $i$ is searching for. Together with $K_p$ , it decides how much attention the view $i$ inpainter should place on the prior view, and finally incorporates the prior view information $V_p$ into view $i$. We will elaborate on these points in the revised version of the paper.
>
> - **2. Discussion of selecting base frame.**
>
> To achieve better generalization, we propose sampling the base frame based on the geometrical centroid of the training camera poses. However, Stable Diffusion occasionally inpaints strange artifacts in the masked region. To mitigate this, we sample $n$ candidate views around the centroid of the camera viewpoints and select the one with the highest similarity votes, automatically avoiding artifacts without human intervention. In our implementation, we used five candidate views, with similarity calculated using perceptual hashing.
>
> We tested our results under different settings (candidate numbers): 3, 5, 7, and 9. The base frame selection algorithm proved to be robust, with our algorithm typically yielding the same base frame. However, another factor influencing this step is the random seed. Setting different seeds causes the 2D inpainting model to produce different results, leading to different base frames being selected. We tested our method under five different seeds, with final scores reported in the table below and additional qualitative results in Figure 2 of the rebuttal PDF. While different seeds affect the appearance of the masked region in the final NeRF, the consistency of multi-view inpainting remains robust, resulting in minimal variance in evaluation scores. We will explicitly clarify this point in the revision.
>
> >| Seed | LPIPS ↓ | MUSIQ ↑ | FID ↓ |
> >|------|---------|---------|-------|
> >| 1    | 0.46    | 46.61   | 264.91|
> >| 2    | 0.44    | 48.04   | 255.29|
> >| 3    | 0.44    | 46.47   | 262.09|
> >| 4    | 0.44    | 45.72   | 261.04|
> >| 5    | 0.46    | 48.65   | 258.50|
> >| *Avg* | *0.45* | *47.10* | *260.37* |
> >| *Std* | *0.01* | *1.21* | *3.657* |
>
> _Table 1: Sensitivity analysis on the prior inpainting results and prior view selection. Results are evaluated on SPIn-NeRF dataset with different random seeds._
>
> - **3. Eq (10) applied to Stage 3.**
>
> Yes, Eq. 10 is the final loss for Stage 3, designed to alleviate noise and inconsistency. Thank you for the reviewer's insight, and I fully understand the confusion. The prior loss $L_\text{prior}$ is an L2 loss first proposed in Stage 2. We propose keeping this term in Stage 3 because we believe the base view does not require patch-based optimization. As discussed in the limitations section, achieving perfect 3D consistency is challenging, so the remaining views may still have subtle inconsistencies. To preserve high-frequency details, we use patch loss instead of the exact-match L2 loss. We regard the base view as a baseline that does not exhibit inconsistency; thus, we do not apply the patch loss to it.
>
> - **4. Discussion of Subset.**
>
> We found that for reconstruction tasks, more views can enhance quality; however, for generation tasks, using the entire set of images introduces unnecessary inconsistencies. Therefore, we propose selecting a subset based on the distribution of camera viewpoints.
>
> We evenly split the viewpoints into 12 groups based on the base view's camera space (evenly 2 on the x and y axes and 3 on the z axis) and select 50 percent frames within each group according to perceptual hashing similarity to the base view. This approach avoids redundant views introducing supervision conflicts, while covering different viewpoints effectively.
>
> We also evaluated our method with different selection percentages, as reported in the table below. The quantitative scores are quite close, indicating minimal differences for most scenes. For one complex scene with high frequencies, setting the percentage too low (0.2) yields artifacts due to insufficient viewpoint coverage, while setting it too high (0.8) introduces appearance conflicts due to variability in inpainted results. These results are visualized in Figure 4 of the rebuttal PDF.
>
> >| Percentage | LPIPS ↓ | MUSIQ ↑ | FID ↓   |
> >|------------|---------|---------|---------|
> >| 0.2        | 0.46    | 45.98   | 265.48  |
> >| 0.4        | *0.44*  | 46.32   | 264.91  |  <!-- Used italics to simulate underlining -->
> >| 0.6        | **0.44**| **47.11** | **261.62** | <!-- Bold for emphasis -->
> >| 0.8        | 0.45    | *46.47* | *263.20* | <!-- Used italics to simulate underlining -->
>
> *Table 2 : Sensitivity analysis on proportion of images selected for the subset.*
>
> Overall, for most scenes, the subset selection algorithm is robust due to the consideration of viewpoints distribution. For extreme cases, careful selection of the percentage might be necessary. However, values between 0.5 and 0.7 remain a reliable choice. We will carefully revise the manuscript to include these details.

---

> > ### Comment · Reviewer_f6e3 · 2024-08-13
> >
> > Thanks for the rebuttal that clarifies my confusion. The additional ablations are helpful. I believe incorporating them will substantially strengthen the paper.
> >
> > Can you further clarify what are the two components in L169 and prompt e in L199?

---

> ### Author Response · Authors · 2024-08-13
>
> Thanks you very much for your feedback.
>
> Regarding L169, the two components described are the input prompt $e$ and the masked image $I_p^{\prime}$ , as detailed in L170.  Figure 2 illustrates the masked image $I_p^{\prime}$. The prompt $e$ is simply set as the description of the background we aim to inpaint, which is scene-dependent.
>
> In L199, the prompt $e$ is defined same as In L169, we employ a unified prompt for each scene.
>
> It's important to note that we did not involve any prompt engineering in our work, and we use the same prompt for each scene across all the methods in the experiments. We will provide further clarification on this point in the revised version.

---

> > ### Comment · Reviewer_f6e3 · 2024-08-14
> >
> > Thanks for the clarification. I'm willing to raise my score and recommend for acceptance.

---

> > > ### Author Response · Authors · 2024-08-14
> > >
> > > Thank you very much for your updated feedback and for considering an increase in your score. We greatly appreciate your support and recommendation for acceptance.

---

### Official Review · Reviewer_uTYn · 2024-07-12

**Soundness:** 2
**Presentation:** 2
**Contribution:** 2
**Rating:** 4
**Confidence:** 4

**Summary:**

This paper presents how to tackle the challenge of 3D object removal by involving 2D diffusion prior. The approach involves pretraining NeRF with inpainting prior and then jointly optimizing it with latent alignment to align feature priors.

**Strengths:**

1. Introducing a 2D diffusion prior as a solution for the 3D inpainting task is an innovative and interesting approach.
2. The incorporation of latent alignment to align feature priors is a promising design.

**Weaknesses:**

1. Although the generated results outperform the listed baseline methods, the paper lacks a comparison with other baselines, such as "Reference-guided controllable inpainting of neural radiance fields" (ICCV 2023)", which achieves good results of novel view synthesis.
2. The presentation of the paper could be improved, for example, it is uncertain how to get the patches in the patch-based loss (Equ. 9).

**Questions:**

1. In Equ. 5, does the first term indicate the use of ground truth images (with the object removed) for supervising the mask region? If so, are there any concerns regarding potential information leakage?
2. Regarding efficiency, does the multi-stage training approach in this paper have a comparison with baseline methods in terms of computational efficiency?

**Limitations:**

The author provides a comprehensive discussion of the broader impacts in the paper.

---

> ### Author Rebuttal · Authors · 2024-08-07
>
> We would like to thank the reviewer for providing these valuable comments.
>
> - **1. Lacks a comparison with other baselines.**
>
> Thank you to the reviewer for pointing this out. Unfortunately, the method "Reference-guided Controllable Inpainting of Neural Radiance Fields" (ICCV 2023) is not open-sourced. However, this method, as stated in the introduction of our paper, operates similarly to the baseline method (InFusion), i.e., using a single reference view to serve the entire scene. These approaches assume that the geometry inferred by the radiance field (NeRF or Gaussian Splatting) is almost accurate, thus relying heavily on the geometric prior.
>
> From the results of InFusion in Figure 4 of the main paper, we can observe that the accuracy of the geometry cannot be guaranteed from scene to scene. For instance, the InFusion excels in the first and third rows but exhibit some disconnection of the primitives in the second and fourth rows. Similarly evidence can be found in our new collected datasets as well, as in rebuttal PDF Figure 1.
>
> These findings highlight the importance of multi-view supervision for 3D generation and inpainting tasks. Consistent multi-view supervision can mitigate the reliance on geometric priors and improve the robustness and accuracy of the inpainting results. By incorporating multi-view supervision, our method still works in the circumstances when such depth/geometry is not accurate, achieving promising results.
>
> - **2. How to get the patches.**
>
> Thank you to the reviewer for the constructive feedback. We will revise the paper to include more complete details in the revision. The patches are uniformly sampled within the bounding box of the mask, and the patch size we used is 256×256. Therefore, only the content within the bounding box (mostly the inpainted area) is being optimized by the patch loss.
>
> - **3. Eq.5 ground truth supervision.**
>
> Thank you to the reviewer for the comment. There is no ground truth supervision for the masked region. We used the official train-test split provided by SPIn-NeRF. In the training set, the masked region contains the unwanted object, while the test set contains the ground truth background in the masked region. We only optimize our radiance field on the training set, ensuring no information leakage. We will explicitly clarify this point in the revision.
>
> - **4. Computational efficiency.**
>
> Thank you to the reviewers for the valuable feedback. We would like to provide a comparison of training time in below table.  InFusion ranks first in efficiency due to the high rendering efficiency of Gaussian splatting and the simplicity of single-view optimization. The three NeRF-based multi-view methods (SPIn-NeRF, NeRFiller, and ours) exhibit similar optimization efficiency. Our method has a slight advantage in efficiency due to the sampled prior image and the relatively consistent multi-view images, which facilitate faster convergence.
>
> >| Method     | Supervision | Training Time (min) |
> >|------------|-------------|---------------------|
> >| SPIn-NeRF  | Multi-view  | 71                  |
> >| NeRFiller  | Multi-view  | 65                  |
> >| InFusion   | Single-view | **17**              |
> >| Ours       | Multi-view  | *49*                |
>
> _Table 1: A comparison of Training Time._

---

> ### Author Response · Authors · 2024-08-13
>
> Dear Reviewer uTYn,
>
> I hope this message finds you well. We are grateful for the time and attention you've already dedicated to reviewing our work. We have submitted a rebuttal addressing the concerns raised, and we would greatly appreciate any additional feedback or comments you might have. As the discussion period is drawing to a close soon, we are eager to know if our rebuttal addresses your concerns or if there is any additional feedback you might have. Thank you very much for your continued support and guidance.
>
> Best regards,
>
> Authors of Submission 4061

---

### Official Review · Reviewer_S5oa · 2024-07-15

**Soundness:** 3
**Presentation:** 3
**Contribution:** 2
**Rating:** 4
**Confidence:** 5

**Summary:**

This paper proposes to solve multi-view or 3D inpainting problem. Given multiple posed views and the masks in each view specifying an object to be removed. It first trains a NeRF on the unmasked regions. To in-paint the masked regions, a seed view is selected and inpainted with a pre-trained diffusion inpainting pipeline. Depth is also predicted for this seed view, together with the RGB value, fused into the NeRF representation. To propagate the features from this view to other views for NeRF supervision, it proposes explicit and implicit feature alignment methods. For the explicit one, it propagates the initial latent noise to another view via a depth/density-based aggregation method.  For the implicit one, it uses a method similar to reference-only control, i.e., extracting intermediate key/value pairs from the seed view and injecting them into other inpainting views via cross-attention.

**Strengths:**

(1) Experiments show that properly propagating anchor-view to other views with methods shows good performance for multi-view consistency. More notably, propagating the initial noise to other views is novel and interesting.

**Weaknesses:**

(1) The proposed method seems much simpler(or even over-simplified) than prior works such as NeRFiller and InFusion. The former uses a synchronized multi-view inpainting technique which is similar to the ILA of this paper. The latter one introduces new diffusion-based depth completion. Both prior methods have used a depth prediction model, and InFusion has shown that using a pre-trained depth model and aligning it with  NeRF/Gaussian depth is sub-optimal.
For qualitative comparisons, there is little or only marginal difference between this paper and InFusion, while InFusion/NeRFillter has shown much more results than only the scenes included in this paper.
I suspect that the result difference is due to the choice of base components, (i.e., inpainting pipeline, depth estimation model, etc.) which can not be considered as the novelty of this paper.

(2) As I understand, the three stages run sequentially. i.e., Train unmasked NeRF(Stage-1) -> Update the NeRF with seed view and depth(Stage-1) -> Update the NeRF with novel inpainted views (guided by the seed view)(Stage2 then 3). Although Equation. (10) shows a total loss function.
- This way, the proposed ELA approach seems over-complicated. After stage 1, there is no NeRF information in the masked region, and there is only one view with depth provided to supervise the masked area. There seems no need to use the NeRF density to propagate the feature from the reference inpainted view to other views. An alternative can be using the depth prior and the epipolar line to aggregate features.

(3) Prior works show both object insertion and removal tasks, while the proposed method seems can only be applied to object removal tasks, as it heavily relies on the initial inpainting and depth estimation result. Moreover, the examples shown in this paper don't include complex structures and only show simple background, inconsistency caused by possible occlusion or partial observation has not been tested. Overall, this paper compares the results with the previous generation's scene completion method in a very limited scenario, which I believe is unfair.

(4) The artifacts are obvious in the result videos, which can be caused by the noise/artifacts in the depth estimation model. More evidence is needed to make this paper stronger.

**Questions:**

See weakness.

**Limitations:**

Limitations addressed, however, the author has not mentioned that the proposed method is only tested in a limited scene case.

---

> ### Author Rebuttal · Authors · 2024-08-07
>
> We appreciate the reviewer for the constructive feedback.  Below we provide the answers to reviewer's concern:
> - **1. Method over-simplified, and marginal difference compared to priors works.**
>
> We would like to argue that while our motivation may appear straightforward, the solution we propose is both non-trivial and novel.
>
> The multi-view inpainter NeRFiller achieves consistent inpainting results by averaging noise predictions derived from multiple iterations. This approach tends to produce smooth results, as demonstrated in Figure 4 of the main paper and Figure 1 of the rebuttal PDF. In contrast, our method explicitly and implicitly aligns latents without sacrificing high-frequency details.
>
> InFusion relies on a single reference view and its depth to represent the entire scene. This reliance becomes problematic when the assumption of accurate depth is violated. As shown in Figure 4 of the main paper, misaligned geometry would lead to disconnected primitives (second and fourth rows). Similar evidence can also be seen in Figure 1 of the rebuttal PDF. By incorporating consistent multi-view supervision, our method remains effective even when depth or geometry is inaccurate, achieving robust and promising results. **This explains why our method shows little difference from InFusion when geometry is accurate** (first and third rows of Figure 4 in the main paper), but excels when the depth is inaccurate.
>
> We further collected nine scenes with manually annotated masks to evaluate the effectiveness of our method. This dataset includes four indoor scenes and five outdoor scenes. We conducted the experiment and evaluation under the same protocol as in the main paper, using the same inpainting pipeline for all methods as mentioned in the paper. The quantitative results are reported in table below and the qualitative results are shown in Figure 1 of the rebuttal PDF. We will include this evaluation with more details in the revised manuscript.
>
> >| Method    | LPIPS ↓ | MUSIQ ↑ | FID ↓  |
> >|-----------|---------|---------|--------|
> >| NeRFiller | 0.68    | 19.43   | 399.19 |
> >| InFusion  | 0.47    | 31.35   | 319.59 |
> >| Ours      | **0.35**  | **37.22** | **250.63** |
>
> _Table1: Quantitative Result on new collected dataset._
>
> In summary, although our motivation is simple, i.e.,using consistently inpainted multi-view images to optimize the radiance field, our approach to achieve consistent results is both novel and effective without relying on any geometric assumptions.
>
>
> - **2. ELA approach seems over-complicated.**
>
> Thank you to the reviewer for providing valuable comments. First, the reviewer's understanding of the three stages is totally correct. However, there are two key reasons why we propose fine-tuning the NeRF and using it as a geometric prior for ELA:
>
>   >(a) After finetuning the NeRF, the geometric is represented by NeRF as a sharp (low variance) unimodal distribution on the ray. Consequently, the aggregated feature remains sharp, preserving the variations in the initial latents.
>
>   >(b) As we discussed in the paper, we empirically found the depth prior inferred by the monocular depth estimator is not perfectly aligned for the NeRF. Fine-tuning the NeRF can also benefit this depth prior. Since NeRF learns relatively certain geometry in the known (unmasked) areas, this geometry constraint can improve the geometry of neighboring inpainted (masked) areas due to their geometric proximity.
>
> We will explain these points in more detail in the revised version of the paper.
>
> - **3. Limited scenario.**
>
> While prior works have been proposed for more general cases, our motivation focuses on a relatively smaller scope, specifically "object removal." As outlined in the paper, we have found that this task is extremely challenging due to the requirement for scene-level generation, which demands consistent multi-view supervision of a scene rather than just an object. Current approaches still exhibit some limitations in this regard.
>
> In contrast, there are many works focusing on 3D object-level generation using multi-view diffusion models. Therefore, we believe that tasks such as object insertion and completion should follow this more promising direction, which we have decided to leave as future work.
>
> However, in response to the reviewer's comments, we have further tested our method on three 3D object completion scenes, which are already included in the nine scenes mentioned in our first response (marginal difference compared to prior works). The results, shown in the rebuttal PDF file Figure 1 (first two rows), indicate that our method still performs promisingly on this complex task. Despite this, we continue to believe that inpainting tasks for objects might be more effectively addressed using multi-view diffusion or video diffusion techniques. We will elaborate on these points in the revised version of the paper.
>
> - **4. Artifacts are obvious in the result video.**
>
> We have carefully examined our implementation of the method and identified that the artifacts are caused by the depth loss used. Initially, we borrowed the loss function proposed in "Depth-supervised NeRF: Fewer Views and Faster Training for Free (CVPR 2022)", which applies KL divergence to the rays' density distribution for sparse view reconstruction. We found that this loss is relatively sensitive and tends to produce more noise in under-constrained areas (with no multi-view supervision). To address this issue, we have replaced our depth loss with a more stable L2 loss, which has largely eliminated the artifacts. The revised implementation now produces significantly cleaner results.

---

> ### Author Response · Authors · 2024-08-13
>
> Dear Reviewer S5oa,
>
> I hope this message finds you well. We are grateful for the time and attention you've already dedicated to reviewing our work. We have submitted a rebuttal addressing the concerns raised, and we would greatly appreciate any additional feedback or comments you might have. As the discussion period is drawing to a close soon, we are eager to know if our rebuttal addresses your concerns or if there is any additional feedback you might have. Thank you very much for your continued support and guidance.
>
> Best regards,
>
> Authors of Submission 4061

---

### Author Rebuttal · Authors · 2024-08-07

We would like to thank all the reviewers for their thoughtful and constructive feedback on our manuscript.

Below, we summarize the modifications made to the manuscript based on your comments.

- To Reviewer S5oa
  - We have clarified the motivation and contribution compared to the priors work.
  - We have explained the rationale behind our ELA approach.
  - We added experiments to further demonstrate the effectiveness of our method, showing promising results for complex tasks like object completion, as shown in Figure 1 of the attachment.
  - The artifacts in the videos have been discussed and resolved.

- To Reviewer uTYn
  - We have clarify the lack of comparison with other baselines.
  - We have detailed the technical process of obtaining patches.
  - We have addressed the concern of potential information leakage.
  - We have provided a comparison of computational efficiency.

- To Reviewer f6e3
  - We have elaborated on the intuition behind ILA with more details.
  - We have discussed our algorithm for selecting the base frame, including a sensitivity analysis and following discussion.
  - We have explained the motivation for retaining the loss of the prior image in Eq. 10.
  - We have discussed our motivation and methodology for selecting subsets, including a sensitivity analysis and following discussion.

- To Reviewer P7tn
  - We have clarify the notation confusion in the paper.
  - We have clarify the compromise of view-dependent effect in ELA.
  - We have conducted four sensitivity analyses on different aspects of our method, providing deeper insights into our approach.

We appreciate the reviewers' insights, which have been invaluable in refining our work.

---

### Decision · Program_Chairs · 2024-09-25

**Decision:**

Accept (poster)

**Comment:**

This paper is about 3D object inpainting using 2D diffusion prior. The paper received mixed scores of two borderline reject, one weak accept, and one accept. The negative reviewers (S5oa, uTYn) have concerns about the details and motivation of the proposed ELA approach, missing comparisons with baselines, and the lack of computational efficiency comparisons. In the rebuttal, the authors provided more detailed descriptions of the proposed method, showed preliminary results on object completion, explained why reference-guided inpainting (ICCV 2023) is not compared (as it is not open-sourced), and provided computational efficiency results. Unfortunately, the reviewers have not responded to these new results. The other two reviewers appreciate the technical novelty of the method and believe that the evaluation is convincing.

Based on the reviews and the author's rebuttal, the AC agrees with the reviewers that the authors' rebuttal largely addresses the initial concerns. The AC believes that this paper has sufficient merits and thus recommends its acceptance.